# A Preliminary Study on the Effect of Hydrogen Gas on Alleviating Early CCl_4_-Induced Chronic Liver Injury in Rats

**DOI:** 10.3390/antiox10121933

**Published:** 2021-12-01

**Authors:** Jianwei Wang, Quancheng Cheng, Jinyu Fang, Huiru Ding, Huaicun Liu, Xuan Fang, Chunhua Chen, Weiguang Zhang

**Affiliations:** Department of Human Anatomy, Histology and Embryology, School of Basic Medical Sciences, Peking University, Beijing 100191, China; wjw@bjmu.edu.cn (J.W.); 2111110029@bjmu.edu.cn (Q.C.); fangjinyu@bjmu.edu.cn (J.F.); dinghuiru@bjmu.edu.cn (H.D.); liuhc@bjmu.edu.cn (H.L.); fangxuan0102@bjmu.edu.cn (X.F.)

**Keywords:** hydrogen gas, chronic liver injury, UCP2, oxidative stress

## Abstract

As a small-molecule reductant substance, hydrogen gas has an obvious antioxidant function. It can selectively neutralize hydroxyl radicals (^•^OH) and peroxynitrite (ONOO^•^) in cells, reducing oxidative stress damage. The purpose of this study was to investigate the effect of hydrogen gas (3%) on early chronic liver injury (CLI) induced by CCl_4_ and to preliminarily explore the protective mechanism of hydrogen gas on hepatocytes by observing the expression of uncoupling protein 2 (UCP2) in liver tissue. Here, 32 rats were divided into four groups: the control group, CCl_4_ group, H_2_ (hydrogen gas) group, and CCl_4_ + H_2_ group. The effect of hydrogen gas on early CLI was observed by serological tests, ELISA, hematoxylin and eosin staining, and oil red O staining. Immunohistochemical staining and Western blotting were used to observe the expression of UCP2 in liver tissues. We found that CCl_4_ can induce significant steatosis in hepatocytes. When the hydrogen gas was inhaled, hepatocyte steatosis was reduced, and the UCP2 expression level in liver tissue was increased. These results suggest that hydrogen gas might upregulate UCP2 expression levels, reduce the generation of intracellular oxygen free radicals, affect lipid metabolism in liver cells, and play a protective role in liver cells.

## 1. Introduction

Chronic liver injury (CLI) is one of the main chronic diseases. With the improvement in living standards, the incidence of CLI has increased over the years and has seriously affected people’s health [1]. CLI has a series of obvious pathological characteristics, including liver steatosis, liver inflammation, cirrhosis, and liver cancer [2]. As an early stage in liver injury, hepatocyte steatosis refers to the occurrence of lipid droplets in the cytoplasm of liver cells beyond the normal physiological range or the appearance of lipid droplets in cells that do not typically contain fat [3]. Hepatocyte steatosis is a reversible disease, and early treatment has a good effect. However, if it is allowed to develop, it may eventually develop into cirrhosis.

Liver injury can be caused by a variety of endogenous and exogenous factors, such as alcohol, drugs, carbon tetrachloride (CCl_4_), and metabolic diseases [4]. CCl_4_ can cause liver cell damage through various mechanisms. Under the catalysis of cytochrome P450, CCl_4_ reacts with surrounding lipids and membrane proteins to generate halane free radicals, lipid free radicals (L^▪^), and protein free radicals (Pr^▪^) [5]. L^▪^ and Pr^▪^ also react with O_2_ to produce lipoalkoxy (LO^▪^), lipoperoxy (LOO^▪^), proteinaloxy (PrO^▪^), and proteinperoxy (PrOO^▪^) [6,7]. In addition, CCl_4_-induced lipid peroxidation products such as 4-HNE also induce the generation of intracellular hydrogen gas peroxide, and peroxides can produce hydroxyl groups under the catalysis of metal ions [8]. CCl_4_ generates various free radicals in a variety of ways. These free radicals can directly attack biological macromolecules in cells such as proteins, DNA, and lipids, causing serious damage to the body.

Hydrogen gas is the lightest known gas. It is colorless and odorless, with only 1/14 of the density of air [9]. Hydrogen gas has strong reducibility and is mainly used as a reducing agent in various industrial reactions to provide electrons to oxidizing substances [10]. In recent experimental studies, hydrogen gas has expanded its original range of applications and entered the medical field as a new antioxidant drug [11,12,13]. A variety of human diseases and the normal aging process of the body are closely related to oxygen free radicals produced by oxidative stress [14]. Hydrogen gas plays a therapeutic role in a variety of oxidative stress diseases, including cerebrovascular diseases [15], type 2 diabetes [16], sepsis [17], and so on.

Uncoupling proteins (UCPs) are a family of ion carriers located in the inner membrane of mitochondria that can mediate proton leakage and reduce the electrochemical gradient along the respiratory chain [18]. In mitochondria, the production of reactive oxide species (ROS) is sensitive to changes in proton dynamic potential energy in the inner membrane, and uncoupling protein-2 (UCP2) can disperse the mitochondrial inner membrane potential and pH gradient, thus reducing the production of ROS. This indicates that UCP2 has the ability to reduce intracellular oxidative stress [19]. However, whether UCP is involved in the oxidation resistance of hydrogen gas is unknown. Therefore, this study used CCl_4_ modeling to investigate the therapeutic effect of hydrogen gas on CLI and preliminarily to explore the protective mechanism of hydrogen gas on hepatocytes by observing the expression of UCP2 in liver tissues.

## 2. Materials and Methods

### 2.1. Rat Grouping and the Early CLI Model Construction

This study was approved by the Ethics Committee for Animal Research Studies at the Peking University Health Science Center (CXK 2016-0010). At least 3 days before the experiment, male Sprague–Dawley rats (250–300 g) were moved to the experimental environment for adaptive feeding, with light and darkness for 12 h each day and free drinking and eating.

Thirty-two rats were randomly divided into four groups: The control group, CCl_4_ group, CCl_4_ + H_2_ group, and H_2_ group. In the control and H_2_ groups, the rats received injections of olive oil on the first day and fifth day (olive oil was injected subcutaneously into the back at a dose of 30 μL/g body weight) and then untreated or treated with hydrogen gas. The rats in the CCl_4_ and CCl_4_ + H_2_ groups received injections of CCl_4_ (105033, Tongguang, China) (40% CCl_4_ in olive oil was injected subcutaneously into the back at a dose of 30 μL/g body weight) on the first day and fifth day. From the first day of the study, rats in the H_2_ and CCl_4_ + H_2_ groups inhaled high concentrations of hydrogen gas (QT13, Jinghui gas, Beijing, China) (3% hydrogen gas + 97% air) for 1 h every day for 7 consecutive days. Tissues were extracted on the eighth day after deep anesthesia with isoflurane.

### 2.2. Serological Index Detection

Blood was collected from the angular vein after deep anesthesia with isoflurane, stored at 4 °C for 30 min, then centrifuged at 4 °C and 2500 rpm for 20 min, and the supernatant was collected. Serological tests of alanine aminotransferase (ALT) and aspartate aminotransferase (AST) were submitted to the Department of Laboratory Animal Science of Peking University Health Science Center, and the serological index test results are shown in Table 1 and Figure 1 as the mean ± SEM.

### 2.3. Histomorphological Observation

The livers were collected and fixed in 10% neutral buffered formalin for histological examination. For the best fixation effect, liver tissue samples of approximately 10 mm × 5 mm × 5 mm were used. Half of the fixed liver tissues in each group were dehydrated by gradient ethanol, transparent and embedded in paraffin. Paraffin-embedded sections were used for hematoxylin and eosin (H&E) staining. The other half of the fixed liver tissue was dehydrated by sucrose and embedded with optimal cutting temperature compound, and the subsequent sections were used for oil red O (ORO) staining. H&E staining and ORO staining were performed using standard procedures, and then, the samples were observed under a microscope (Olympus, Tokyo, Japan). For immunohistochemistry, liver tissue sections were incubated overnight with the primary antibody, followed by treatment with HistostainTM-plus kits (ZSGB-BIO, Beijing, China). The final stage of immunohistochemical staining was to display the antigen–antibody complexes using a DAB substrate kit. The primary antibody used in our study was anti-UCP2 (1:500, HPA075202, Sigma, St. Louis, MO, USA). Percentages of positive particles or areas visualized by microscopy were quantified by automated counting (ImageJ) on two or three fields of view selected randomly from each section. Two sections were selected from each group. The image capture was performed using a Leica DM5000 microscope.

### 2.4. Western Blotting

The liver tissues were lysed in cold RIPA buffer (Applygen, Beijing, China) for 15 min. The lysate was centrifuged at 12,000 rpm for 10 min at 4 °C. A 30 μL volume of protein was separated by SDS–PAGE gel. Then, the separated proteins were transferred to PVDF membranes and blocked in 5% skim milk (Harveybio, China) dissolved in Tris-buffered saline-Tween-20 (TBST) (Sinodetech, China) for 1 h at room temperature. The membrane was incubated with antibodies of goat anti-UCP2 (bs-20750R, Bioss, Beijing, China) and mouse anti-β actin (A2228, Sigma, St. Louis, MO, USA) overnight at 4 °C. Then, the membrane was incubated with horseradish peroxidase-conjugated anti-goat or anti-mouse IgG antibody (A5420, AP124P, Sigma, St. Louis, MO, USA) for 1 h at room temperature. The membrane was visualized using a superenhanced chemiluminescence reagent (ECL).

### 2.5. Enzyme-Linked Immunosorbent Assay (ELISA)

ELISA of interleukin-6 (IL-6), malondialdehyde (MDA), and superoxide dismutase (SOD) was performed according to the relevant kit instructions (Dogesce, Beijing, China).

### 2.6. Statistical Analysis

Data were expressed as the mean ± SEM. GraphPad Prism 6.0 (GraphPad, San Diego, CA, USA) was used for data analysis and statistical image processing. Multiple comparisons were statistically analyzed with one-way analysis of variance (ANOVA) followed by Tukey’s multiple comparison post hoc analysis. *p* < 0.05 was considered significant.

## 3. Results

### 3.1. Hydrogen Gas Inhalation Improved Liver Function in CLI

The serum ALT and AST levels in the CCl_4_ group were significantly increased compared with those in the control group (*p* < 0.05, Figure 1a,b, Table 1). After hydrogen gas inhalation, the ALT and AST levels in the CCl_4_ + H_2_ group were significantly lower than those in the CCl_4_ group (*p* < 0.05, Figure 1a,b, Table 1). There was no significant difference between the H_2_ group and the control group (*p* ≥ 0.05, Figure 1a,b, Table 1). These results indicated that hydrogen gas inhalation could improve liver function in CLI. However, hydrogen gas did not affect the liver function of healthy rats.

### 3.2. Hydrogen Gas Inhalation Reduced CCl_4_-Induced Inflammation

The expression of IL-6, an inflammatory factor, in the CCl_4_ group was significantly higher than that in the control group (*p* < 0.05, Figure 2). After inhaling hydrogen, this index decreased significantly (*p* < 0.05, Figure 2), but there was no difference between the H_2_ group and the control group (*p* ≥ 0.05, Figure 2). These results confirmed that hydrogen gas inhalation could reduce the inflammation induced by CCl_4_.

### 3.3. Hydrogen Gas Inhalation Reduced CCl_4_-Induced Hepatocyte Steatosis

H&E staining of liver tissue from the control group showed that the structure of the hepatic lobules was clear and that the hepatocytes were radially arranged around the central vein in normal hepatic lobules with clear nuclei and uniform cytoplasm (Figure 3a,b). However, in liver tissue from the CCl_4_ group, the lobular structure was disordered, fat degeneration in liver cells was observed, and the nucleus was pushed to one side (Figure 3c,d). The hepatic lobule structure of the H_2_ group was essentially the same as that of the control group (Figure 3g,h). However, the hepatic lobule structure in the CCl_4_ + H_2_ group was clearer than that in the CCl_4_ group, and more orderly liver cells had a less vacuolar structure (Figure 3e,f). Statistical data showed that the CCl_4_ group had a higher nonalcoholic steatohepatitis (NASH) activity score than the control group, but this score was significantly lower in the CCl_4_ + H_2_ group than that in the CCl_4_ group (Figure 4).

ORO staining showed that there was no red droplet deposition in the cytoplasm in the control group (Figure 5a,b). However, numerous bright red droplets of different sizes were observed in the CCl_4_ group. The nuclei, which were stained blue, were forced towards the edge by red particles (Figure 5c,d), and there was no red droplet deposition in the cytoplasm of liver cells in the H_2_ group (Figure 5g,h). The CCl_4_ + H_2_ group showed a significant reduction in red lipid droplets compared to the CCl_4_ group (Figure 5e,f). Statistical data showed that the ORO positive area in the CCl_4_ group was significantly increased compared with the control group, but the CCl_4_ + H_2_ group showed a downward trend (Figure 6).

These results suggest that hydrogen gas inhalation can significantly improve the structure of hepatic lobules and reduce the accumulation of fat in hepatic cells. However, it has no effect on the morphology of the liver in healthy rats.

### 3.4. Hydrogen Gas Inhalation Reduced CCl_4_-Induced Oxidative Stress

MDA, an end product of membrane lipid peroxidation, reflects the degree of tissue peroxidation damage [20]. The CCl_4_ group showed significantly higher levels than the control group (*p* < 0.05, Figure 7a). The level of MDA in the CCl_4_ + H_2_ group was decreased compared with that in the CCl_4_ group, and the difference was statistically significant (*p* < 0.05, Figure 7a). SOD, a major antioxidant enzyme in the body’s oxidative stress response, reflects the ability of tissue to resist oxidative damage [20]. The change in enzyme activities was shown in Figure 7b. The activity in the CCl_4_ group was significantly lower than that in the control group (*p* < 0.05). The enzyme activity of SOD in the CCl_4_ + H_2_ group was higher than that in the CCl_4_ group (*p* < 0.05, Figure 7b). These results suggest that CCl_4_ can induce oxidative stress in rat liver.

### 3.5. Inha lation of Hydrogen Gas Increased UCP2 Expression in Hepatocytes

Immunohistochemical staining showed different amounts of brownish-yellow UCP2 particles in the liver tissues of the control group, CCl_4_ group, H_2_ group, and CCl_4_ + H_2_ group. A small number of UCP2-positive cells were found in the liver tissues of the control group and CCl_4_ group (Figure 8a–d). The number of UCP2-positive cells in the H_2_ group was significantly higher than that in the control group, and the brown-yellow substances were evenly distributed in the cytoplasm (*p* < 0.05, Figure 8a,b,g,h and Figure 9). UCP2 was diffusely expressed in the liver tissues of the CCl_4_ + H_2_ group. It was mostly located on the side of the cell membrane, presenting a dotted and linear distribution (Figure 8e,f). There were significantly more UCP2 positive particles in the CCl_4_ + H_2_ group than those in the CCl_4_ group (*p* < 0.05, Figure 9).

Western blotting results showed that UCP2 expression (UCP2/β-action relative optical density) in liver tissues of the control group, CCl_4_ group, CCl_4_ + H_2_ group, and H_2_ group was 0.21 ± 0.17, 0.46 ± 0.22, 0.63 ± 0.19, and 0.68 ± 0.15, respectively. The H_2_ and CCl_4_ + H_2_ groups were significantly different from the control group (*p* < 0.05, Figure 10). These results suggest that hydrogen gas may promote the expression of UCP2 in hepatocytes. 

## 4. Discussion

CLI has the characteristics of a series of obvious pathological changes, including hepatocyte steatosis, liver inflammation, cirrhosis, and liver cancer, among which hepatocyte steatosis is an early symptom of liver injury [2]. Early treatment can have a good effect, but if left untreated, it will eventually become irreversible liver cirrhosis. CCl_4_ can increase the synthesis of fatty acids and triglycerides and promote lipid peroxidation. Thus, it can reduce the secretion of VLDL in liver cells and lead to lipid transport disorders [7]. Therefore, we established an early CLI model by subcutaneous injection of CCl_4_ into the back. Both H&E staining and ORO staining showed obvious damage to liver lobules and fat accumulation in liver cells, which demonstrated the success of the model construction. Compared with most current methods of CCl_4_-induced CLI [21], this experiment shortened the modeling time. Therefore, this model produced steatosis without fibrosis, to study the therapeutic effect of hydrogen gas on early CLI.

Many studies have shown that hydrogen gas, as a new type of antioxidant, plays a therapeutic role in various oxidative stress diseases [22,23,24]. Hydrogen gas can exert a protective effect against liver ischemia–reperfusion combined with resection injury [25]. In our study, inhalation of a high concentration of hydrogen gas (3%) was used to treat CCl_4_-induced hepatic steatosis. The results showed that inhaling a high concentration of hydrogen gas could significantly reduce the content of ALT and AST in serum and improve hepatic steatosis. These results are probably related to its antioxidant effect. In cultured cells, hydrogen gas reacts directly and selectively with the most oxidizing oxygen radical, ^▪^OH [12]. Hydrogen gas also reacts with ONOO^▪^ in cell-free systems. Two free radicals, ^▪^OH and ONOO^▪^, play a key role in CCl_4_-induced liver injury [26]. On the one hand, hydrogen gas effectively combats the damage caused by oxidative stress in the body. On the other hand, according to the study of Ohsawa I et al. [12], hydrogen gas does not react with some oxygen radicals with weak oxidation capacity in the body, such as O_2_^▪−^, H_2_O_2_^▪−^, NO^▪^, NAD^+^, FAD^+^, and oxidized vitamin C. These oxidizing free radicals have very important signal transduction functions in the cell. Hydrogen gas rarely affects the normal metabolism of the body and normal physiological parameters of the human body, such as body temperature, blood pressure, pH of body fluid, and blood oxygen partial pressure. We also found that inhalation of hydrogen gas did not affect liver function or liver morphology in healthy rats.

UCPs are a family of ionophores located in the inner membrane of mitochondria. They mediate proton leakage and thus reduce the electrochemical gradient along the respiratory chain [18]. UCP2 is closely related to the production of oxygen free radicals. In mitochondria, ROS production is sensitive to changes in proton dynamic potential energy in the inner membrane, while UCP2 can disperse the mitochondrial inner membrane potential and pH gradient to reduce ROS production, indicating that UCP2 has the ability to reduce intracellular oxidative stress [18]. UCP2 is involved in a wide range of physiological and pathological processes in vivo, including cell protection [27], immune cell regulation [28], and glucose sensitivity regulation [29] in the brain and pancreas. A growing number of studies have shown that UCP2 plays a protective role in various tissues by reducing the level of oxidative stress in cells, which is an important indicator of the potential relationship between hydrogen gas and UCP2 [28,30,31]. In our study, hepatocyte steatosis induced by CCl_4_ was treated with hydrogen gas. UCP2 expression was observed in all groups. However, the expression of UCP2 in the H_2_ group was significantly enhanced compared with that in the groups without hydrogen gas, suggesting that hydrogen gas may promote the expression of UCP2 in cells. UCP2, as a phosphate dissolve coupling protein, can regulate ATP production in vivo and affect the energy metabolism system of cells [32]. The effects of elevated UCP2 are a double-edged sword. On the one hand, it can relieve oxidative stress in the body and play a protective role in the body. On the other hand, although the high expression of UCP2 does not affect the liver function and liver morphology of healthy rats in our study, it should not be ignored that UCP2 can reduce the potential level of the respiratory chain, reduce the power of electron transfer, and lead to reduced ATP production.

## 5. Conclusions

Hydrogen gas can upregulate UCP2 expression levels, reduce the generation of intracellular oxygen free radicals, affect lipid metabolism in liver cells, and play a protective role in liver cells.

## Figures and Tables

**Figure 1 antioxidants-10-01933-f001:**
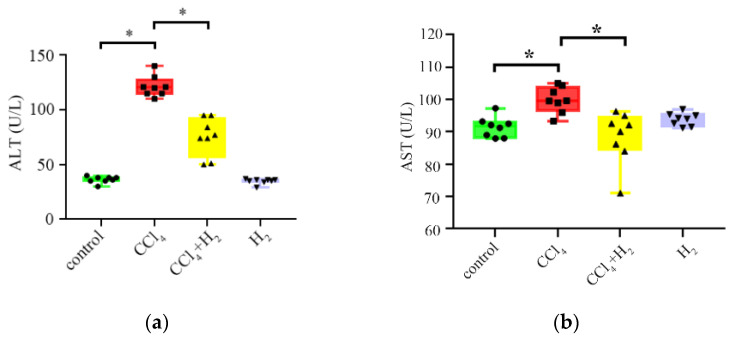
Changes of liver injury index in serum: (**a**) alanine aminotransferase (ALT) level; (**b**) aspartate aminotransferase (AST) level. N = 8, * *p* < 0.05.

**Figure 2 antioxidants-10-01933-f002:**
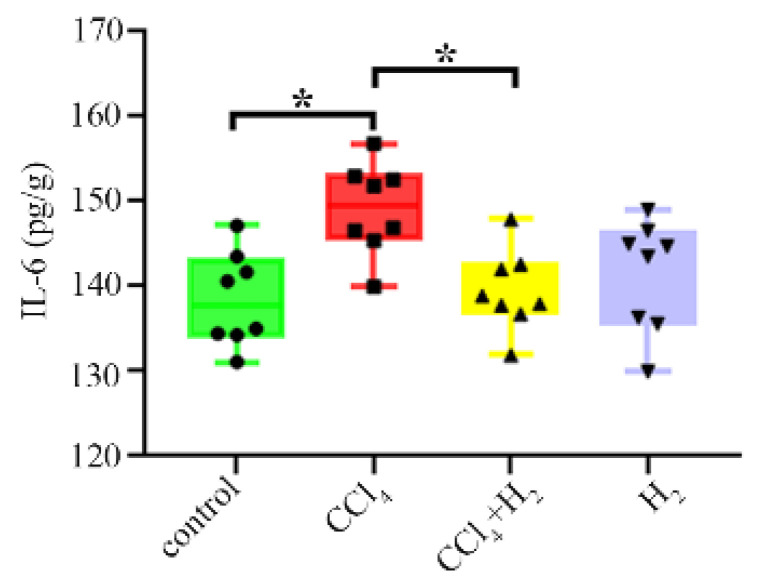
Change in interleukin-6 (IL-6) in livers. N = 8, * *p* < 0.05.

**Figure 3 antioxidants-10-01933-f003:**
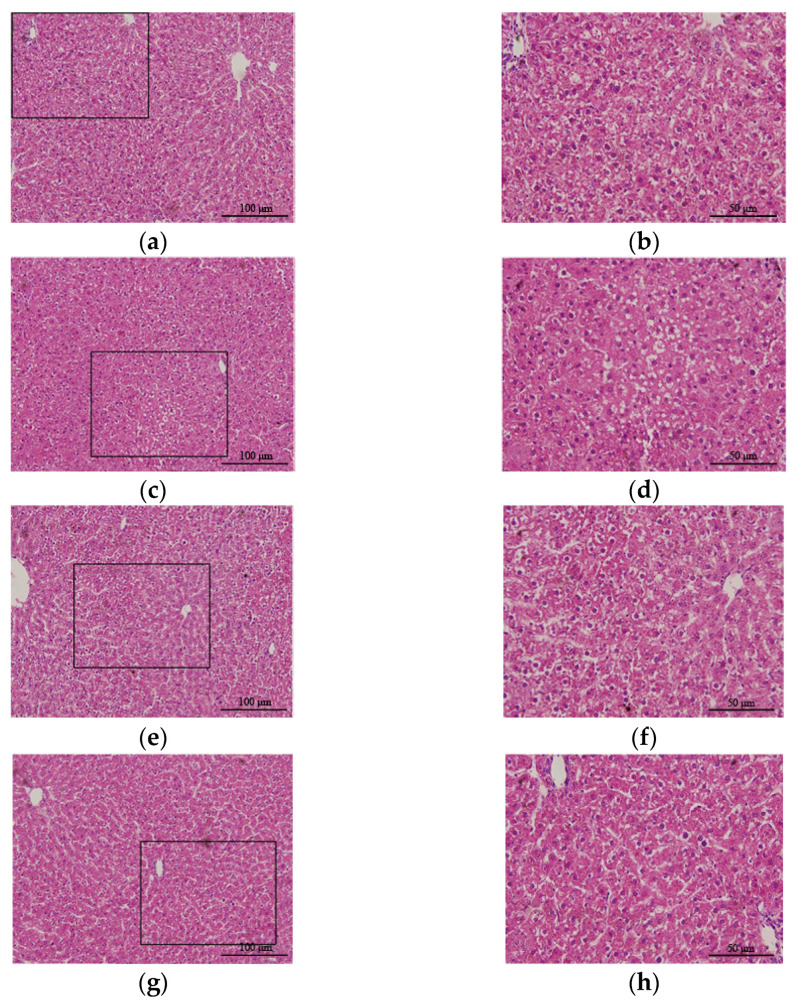
Hematoxylin and eosin (H&E) staining of a paraffin section of liver tissue. (**a**,**b**) Liver tissue of control group rats. Hepatic lobule structure was normal. (**c**,**d**) Liver tissue of CCl_4_ group rats. Hepatic lobule disorder with reduced or absent hepatic sinuses. Vacuolar structures could be seen in liver cells. (**e**,**f**) Liver tissue of CCl_4_ + H_2_ group rats. The hepatic lobule structure was basically normal, and no vacuolar structure was found in the liver cells. (**g**,**h**) Liver tissue of H_2_ group rats. Hepatic lobule structure was normal. (**a**,**c**,**e**,**g**) Fields of vision at 40× magnification. (**b**,**d**,**f**,**h**) Higher magnifications of the area outlined in (**a**,**c**,**e**,**g**). The long scale bar refers to 100 μm. The short scale bar refers to 50 μm.

**Figure 4 antioxidants-10-01933-f004:**
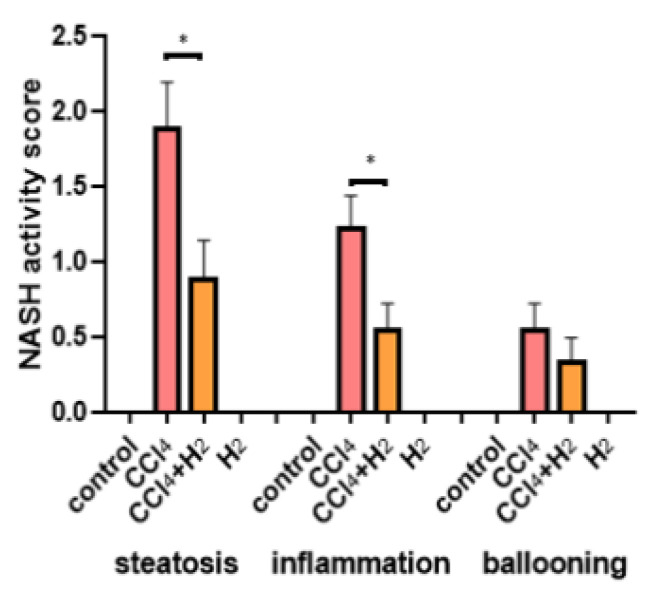
Nonalcoholic steatohepatitis (NASH) activity score. N = 8, * *p* < 0.05.

**Figure 5 antioxidants-10-01933-f005:**
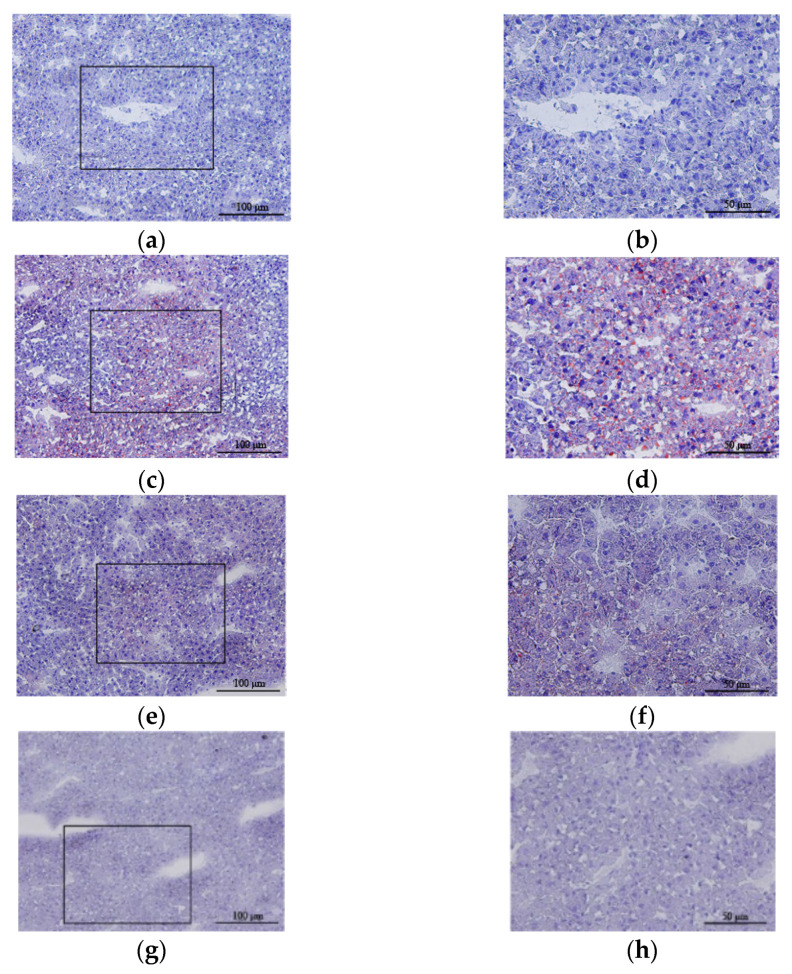
Oil red O (ORO) staining of a frozen section of liver tissue. (**a**,**b**) Liver tissue of control group rats. No red droplet was observed in liver cells. (**c**,**d**) Liver tissue of CCl_4_ group rats. There were a large number of red droplet deposits in the liver cells, mostly clustered around the portal area. (**e**,**f**) Liver tissue of CCl_4_ + H_2_ group rats. There were few red droplets in liver cells. (**g**,**h**) Liver tissue of H_2_ group rats. No red droplet was observed in liver cells. (**a**,**c**,**e**,**g**) Fields of vision at 40× magnification. (**b**,**d**,**f**,**h**) Higher magnifications of the area outlined in (**a**,**c**,**e**,**g**). The long scale bar refers to 100 μm. The short scale bar refers to 50 μm.

**Figure 6 antioxidants-10-01933-f006:**
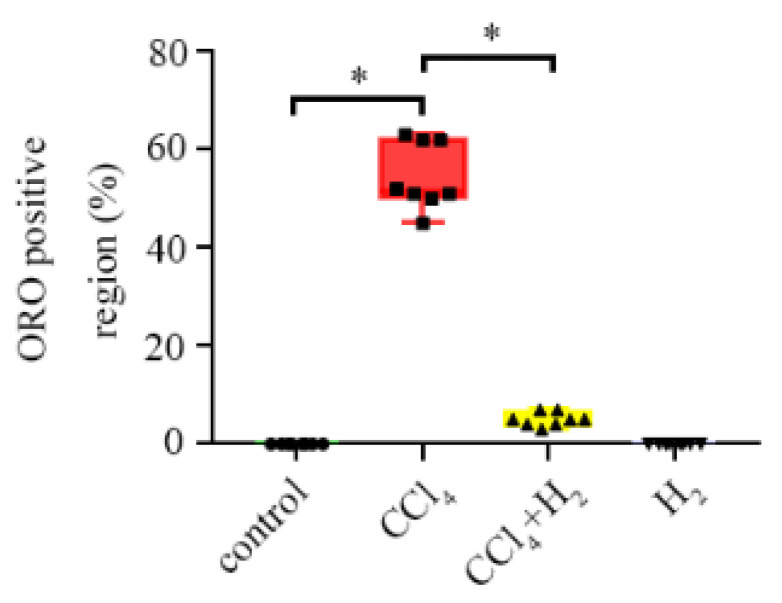
ORO positive area rate. N = 8, * *p* < 0.05.

**Figure 7 antioxidants-10-01933-f007:**
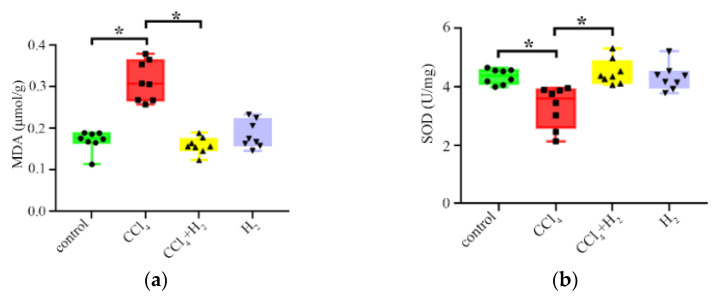
Changes in the expression of oxidative stress-related factors. (**a**) Change in malondialdehyde (MDA) in livers. (**b**) Change in superoxide dismutase (SOD) in livers. N = 8, * *p* < 0.05.

**Figure 8 antioxidants-10-01933-f008:**
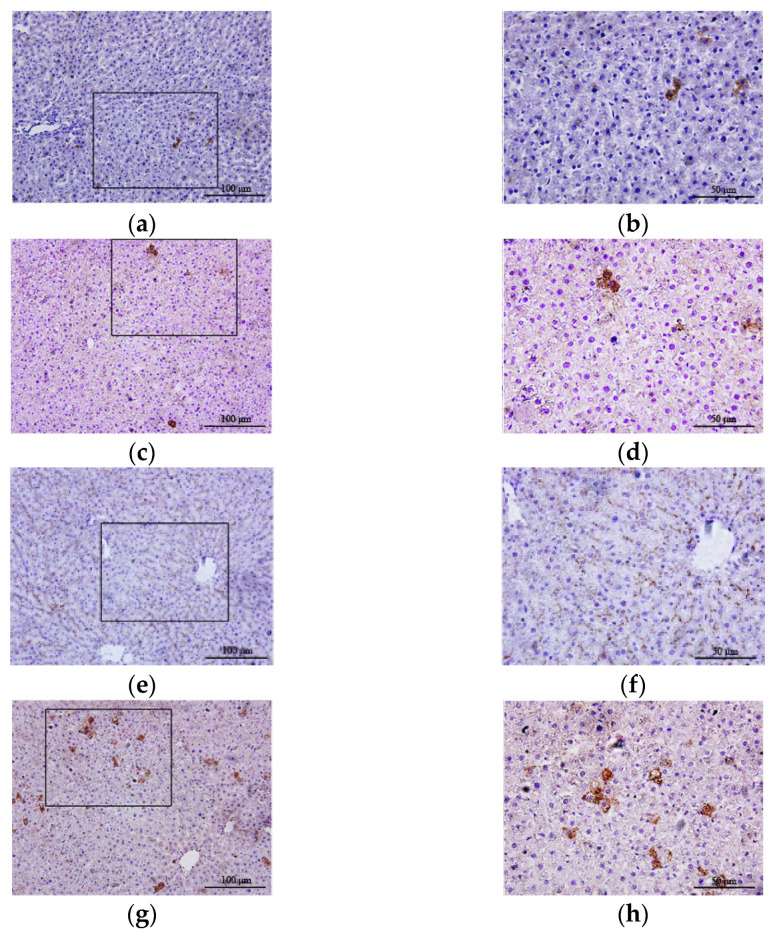
Changes in the expression of uncoupling protein 2 (UCP2) were observed by immunohistochemistry. (**a**,**b**) The expression of UCP2 in liver tissue of control group rats. A small number of UCP2-positive cells were brownish yellow in the cytoplasm. (**c**,**d**) The expression of UCP2 in liver tissue of CCl_4_ group rats. A small number of UCP2 positive cells were brownish yellow in the cytoplasm. (**e**,**f**) The expression of UCP2 in liver tissue of CCl_4_ + H_2_ group rats. UCP2 was diffusely expressed in liver cells, and the staining sites were mostly located on the cell membrane. (**g**,**h**) The expression of UCP2 in liver tissue of H_2_ group rats. The number of UCP2 positive cells with brownish-yellow staining in the cytoplasm increased significantly. (**a**,**c**,**e**,**g**) Fields of vision at 40× magnification. (**b**,**d**,**f**,**h**) Higher magnifications of the area outlined in (**a**,**c**,**e**,**g**). The long scale bar refers to 100 μm. The short scale bar refers to 50 μm.

**Figure 9 antioxidants-10-01933-f009:**
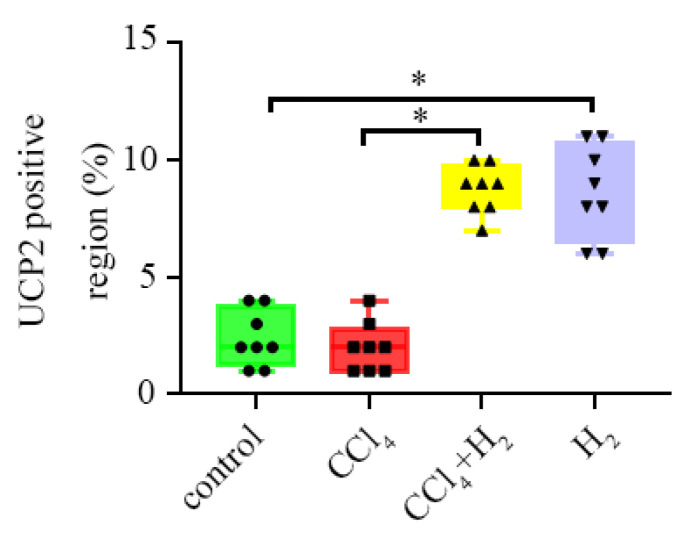
UCP2-positive area rate. N = 8, * *p* < 0.05.

**Figure 10 antioxidants-10-01933-f010:**
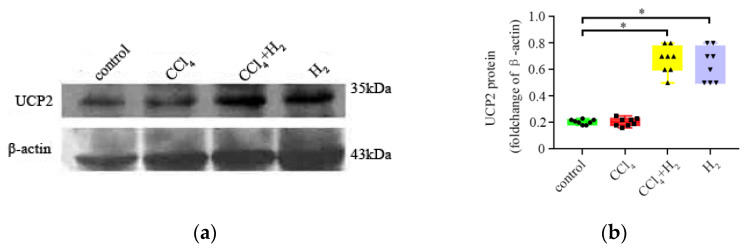
Expression changes of UCP2 (Western blotting). (**a**) Representative western blotting image of UCP2. (**b**) Statistical results of western blotting. N = 8, * *p* < 0.05.

**Table 1 antioxidants-10-01933-t001:** Statistical data of serum ALT and AST.

	Control	CCl_4_	CCl_4_ + H_2_	H_2_
ALT (U/L)	36.25 ± 1.08	121.50 ± 3.37 *	75.00 ± 6.11 ^#^	34.75 ± 0.88
AST (U/L)	91.43 ± 1.11	99.88 ± 1.41 *	88.41 ± 2.89 ^#^	93.88 ± 0.70

* *p* < 0.05 vs. control group. ^#^
*p* < 0.05 vs. CCl_4_ group.

## Data Availability

The data presented in this study are available in this manuscript.

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
