# Peer review of "A Preliminary Study on the Effect of Hydrogen Gas on Alleviating Early CCl4-Induced Chronic Liver Injury in Rats"

_antioxidants, 2021, doi:10.3390/antiox10121933_

Round 1

Reviewer 1 Report

The Authors evaluated impact of hydrogen gas on CCl4-induced liver injury in the stage of steatosis, but not yet fibrosis. Liver disorder, leading finally to irreversible liver cirrhosis, is a great clinical problem. In this aspect, searching for new, not-toxic strategies of prevention or reversion of early injuries, is very important. However, the presented study requires important modifications and corrections before publication.

Title: consider adding “preliminary study”, as suggested in the abstract.

Materials and methods:

  1. in the description is mentioned that the study is performed on mice, but from the main title and further description of methods and results it seems to be on rats – please correct this – lines 68, 70, 73
  2. please add information about the origin of CCl4 and hydrogen gas (manufacturer)
  3. line 80 – it is not clear how exactly rats were exposed to hydrogen gas – 7 consecutive days from the first day of the study, from the 5th days of the study (second CCl4 injection) or maybe 7 days before the first injection of CCl4? – please specify
  4. line 87 – please add in which table and graph – add number
  5. please provide the details of the manufacturers of the microscope, reagents used for Western blot analysis, software SPSS 22 and GraphPad Prism

Results

  1. consider not to double results on figures and tables (Fig. 1 and Tab. 1)
  2. legend to Fig. 2, Fig.4, and Fig. 6 – consider adding the magnifications of images not only bar scale
  3. line 173 – p ≤05 or p = NS

Discussion

  1. line 238 – “in rat liver transplantation major and liver resection” – statement not clear, please rewrite
  2. lines 238-239 – “In this study” – suggests the study cited earlier – maybe better “In our study”
  3. line 266 – citations needed after “A growing number of studies have shown that UCP2 plays a protective role in various tissues by reducing the level of oxidative stress in cells, which is an important indicator of the potential relationship between hydrogen gas and UCP2.”

References

  1. all references require careful check – many inaccuracies e.g.:
    1. 24 – from text indicates study on rats (line 238) but the ref. describes study on miniature pigs
    2. 12 – in the text is “study of Ikuroh O et al. (line 247) but the ref. 12 is Ohsawa I et al. (lines 316-317)

Check the manuscript with native speaker – some typing and grammar errors should be corrected (e.g., line 12 – effect of hydrogen; line 52 – aging; line 74 – treatment; line 215 – UCP2/β-actin; line 247 – et al.)

Reviewer 2 Report

In this report. Wang et al investigated the effects of hydrogen inhalation against CCl4 induced chronic liver damage. They observed that hydrogen inhalation alleviated serological ALT level increase and liver steatosis that commonly associated with CCl4 induced liver injury. Further, they showed that hydrogen gas treatment upregulated UCP2 expression in liver tissue, which suggested that hydrogen gas might upregulate UCP2 expression levels to quench the

intracellular oxygen free radicals in liver cells and protect Hepatocyte from CCl4 induced damage. Although these observations are relative novel, some issues and concerns need to be addressed to confirm the findings:

  1. In addition to ALT level, other serological test for liver function such as AST, Albumin and total bilirubin should be examined as well.
  2. H&E staining was performed to show the lobular structure disorder in CCl4 treated rat liver tissue (figure 2). However, other immunohistochemistry staining and histological analysis need to be performed to examine the death of hepatocyte and subsequent liver fibrosis in the liver tissues from CCl4 treated or CCl4 in combination of hydrogen inhalation treated animals.
  3. Lacking of mechanism connection between hydrogen treatment and upregulation of UCP2 in liver tissues.
  4. There was no direct experimental evidence to demonstrate that hydrogen treatment reduced the CCl4induced free radical or lipid peroxidation products such as 4-HNE in liver tissue.

Reviewer 3 Report

Major changes are needed, there are serious errors in the work.

Round 2

Reviewer 3 Report

There has been an important change in the article since the previous version, improving considerably its quality, the changes in the naming of the groups improves the understanding of the work and new studies are recommended to deepen in the conclusion raised by the authors.

Author Response

We appreciated the comments. On this basis, we have detected AST, IL-6 and molecules responding to oxidative stress (MDA and SOD) to further improve the experimental results and make the experimental conclusions more reliable.